# The Intracellular and Secreted Sides of Osteopontin and Their Putative Physiopathological Roles

**DOI:** 10.3390/ijms24032942

**Published:** 2023-02-02

**Authors:** Ana Clara Santos da Fonseca Bastos, Amanda Vitória Pampolha Gomes, Gabriela Ribeiro Silva, Mariana Emerenciano, Luciana Bueno Ferreira, Etel Rodrigues Pereira Gimba

**Affiliations:** 1Grupo de Hemato-Oncologia Molecular, Coordenação de Pesquisa, Instituto Nacional de Câncer, Rio de Janeiro 20230-130, CEP, Brazil; 2Programa de Pós-Graduação Stricto Sensu em Oncologia, Instituto Nacional de Câncer, Rio de Janeiro 20231-050, CEP, Brazil; 3Programa de Carcinogênese Molecular, Coordenação de Pesquisa e Inovação, Instituto Nacional de Câncer (INCA), Rio de Janeiro 20230-130, CEP, Brazil; 4Programa de Pós-Graduação em Ciências Biomédicas, Fisiologia e Farmacologia, Instituto Biomédico, Niterói 24210-130, CEP, Brazil; 5Departamento de Ciências da Natureza, Universidade Federal Fluminense, Rio das Ostras 28880-000, CEP, Brazil

**Keywords:** osteopontin, secreted osteopontin, intracellular osteopontin, physio-pathological roles, structure, osteopontin isoforms

## Abstract

Classically, osteopontin (OPN) has been described as a secreted glycophosprotein. Indeed, most data concerning its physiological and pathological roles are mainly related to the secreted OPN (sOPN). However, there are several instances in which intracellular OPN (iOPN) has been described, presenting some specific roles in distinct experimental models, such as in the immune system, cancer cells, and neurological disorders. We herein aimed to highlight and discuss some of these secreted and intracellular roles of OPN and their putative clinical and biological impacts. Moreover, by consolidating data from the OPN protein database, we also analyzed the occurrence of signal peptide (SP) sequences and putative subcellular localization, especially concerning currently known OPN splicing variants (OPN-SV). Comprehending the roles of OPN in its distinct cellular and tissue environments may provide data regarding the additional applications of this protein as biomarkers and targets for therapeutic purposes, besides further describing its pleiotropic roles.

## 1. Introduction

Osteopontin (OPN), encoded by the secreted phosphoprotein 1 (SPP1) gene, also known as early T-lymphocyte activation 1 protein (ETA1), is a member of the small integrin-binding ligand N-linked glycoprotein (SIBLING) family. OPN is an acid glycosylated phosphoprotein expressed in several tissues and is involved in many physiologic and pathologic processes, including vascularization, cell regeneration, calcification, inflammatory and immune-modulating disorders, and neurologic disorders, as well as in cancer [1]. It has also been hypothesized that diabetes-induced OPN and its target product Furin result in worse outcomes in diabetic patients with SARS-CoV-2 infection, owing to the roles of these proteins in promoting viral infection and increasing metabolic dysfunction [2].

This glycophosphoprotein is synthesized by many cell types, such as osteoclasts, osteoblasts, epithelial, endothelial, neuronal, and immune cells (T cells, NK cells, macrophages, and Kupffer cells), and is constitutively expressed in several tissues (kidney, breast, brain, skin, bone, bone marrow, and bladder) and biological fluids, such as plasma, urine, milk, and bile [3].

The name osteopontin, in which “osteo” means bone and “pontin” means bridge, is associated with its function of linking bone cells to bone extracellular matrix. This protein is a major component of the bone matrix and is expressed at different stages of bone formation, remodeling, and resorption. In bone remodeling, OPN generally acts by inhibiting mineralization by promoting osteoclast differentiation and activity [4], although it can also behave as a stimulatory factor for mineralization [5].

The SPP1 gene is located on chromosome 4, locus q22.1 (Figure 1A). OPN expression is influenced by genetic polymorphisms in its promoter region, and a large number of polymorphisms can be found along the gene and are associated with cancer and autoimmune diseases. It has been reported 310 variant sequences of the human SPP1 gene. Approximately 10 of them correspond to short deletion and insertion polymorphisms, and the remaining 300 correspond to single nucleotide polymorphisms [6].

This protein consists of 314 amino acid residues with molecular weights ranging from 41 to 75 kDa. In detail, the SPP1 gene comprises seven exons, exon 1 untranslated, while exons 2–7 contain the coding sequences (Figure 1A). Exon 2 codes for the signal peptide (SP) and two amino acids of the mature protein. Exon 3 codes the sequences of serine phosphorylation, while exon 4 codes for a proline-rich region and a transglutaminase site, and exon 5 codes for another protein phosphorylation site. Finally, exons 6 and 7 encode more than 80% of the OPN protein. This region carries an Arg-Gly-Asp (RGD) sequence that interacts with αvβ1, αvβ3, and αvβ5 integrins receptors. The adjacent Ser-Val-Val-Tyr-Gly-Leu-Arg (SVVYGLR) sequence domain interacts with α9β1, α4β1, and α4β7 integrins, which are exposed in response to thrombin cleavage. Thrombin cleavage generates an N-terminal fragment that comprises the integrin binding site and gives rise to a C-terminal fragment containing calcium and heparin-binding sites, which interacts with the cluster of differentiation 44 (CD44) isoforms, such as CD44v3, CD44v6, and CD44v7 [7,8] (Figure 1A and Figure 2). In addition to thrombin cleavage, OPN can also be cleavaged by matrix metalloproteinases (MMP)-3 and MMP-7, plasmin, and cathepsin D, rendering the above domains more accessible, then enhancing OPN binding efficiency. These enzymatic cleavages generate OPN variants that retain their activity or acquire additional functions [9,10].

**Figure 1 ijms-24-02942-f001:**
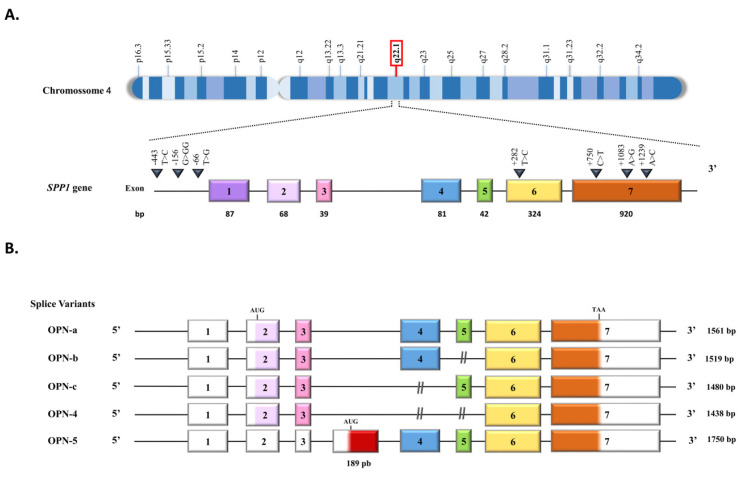
Schematic representation of the OPN gene and its transcripts. (**A**) OPN encoding gene located on human chromosome 4q22.1. The gene is 14,762 bp long and contains 7 exons. (**B**) OPN splice variants. The OPN primary transcript is processed by the alternative splicing resulting in at least five variants: OPN-a contains 7 exons and is known as the canonical isoform; OPN-b lacks exon 5 (green box); OPN-c lacks exon 4 (blue box); OPN-4 lacks both exon 4 and exon 5 (blue and green boxes); and OPN-5, which has an extra exon (red box), resulting from the retention of an intronic portion between exon 3 and exon 4, being the longest OPN splice variant. OPN-5 starts its translation at an alternative site located in the extra exon. (**C**) OPN-5 subvariants. OPN-5a is the OPN-5, as described above; OPN-5b has the extra truncated exon; OPN-5c has an additional 9 bp in the 3’ region in the extra exon (red box surrounded by a bolded black line); OPN-5d has an additional 9 bp in the 3’ region in the extra exon and deletion of exon 5; OPN-5e lacks exon 5. This Figure was partially based on information provided in previous reports [9,11,12].

OPN is a negatively charged protein due to the presence of multiple aspartate residues and displays a secondary structure with 8 α-helices and 6 β-sheets [13]. It is also an intrinsically disordered protein, displaying simultaneously sample extended, random coil-like conformations and stable, cooperatively folded conformations [14,15].

OPN displays several isoforms generated by alternative splicing [16], alternative translation, and post-translational mechanisms [17], such as serine/threonine phosphorylation, glycosylation, tyrosine sulfation, and proteolytic cleavage. These modifications are cell-type specific and depend on pathophysiological mechanisms that affect OPN structure, protein–protein interactions, and functions. The sum of all these variants comprehends the total OPN (tOPN), generally named only as OPN in most reports.

The OPN primary transcript is subject to alternative splicing, generating at least five splice variants (OPN-SV) named OPN-a, OPN-b, OPN-c, OPN-4, and OPN-5 [18] (Figure 2). OPNa is the full-length variant; OPN-b lacks exon 5, while OPNc lacks exon 4. In OPN-4, both exons 4 and 5 are deleted, while OPN-5 contains an extra exon generated from the retention of a portion of intron 3 of the canonical isoform [18]. Once OPN-5 has a different start codon, it is the largest OPN transcript. More recently, four additional splice variants have been described for OPN-5, which were named OPN-5b, OPN-5c, OPN-5d, and OPN-5e [12] (Figure 2). The roles of OPN-a, OPN-b, and OPN-c have been extensively investigated, mainly in human tumor cells, in which these splice variants are aberrantly expressed and present tumor and tissue-specific roles [19]. We also recently found that OPN-4 and OPN-5 are co-expressed in several human cancer cell lines and tumor tissues [17], further evidencing their roles in cancer biology, as has been previously found in several reports regarding OPN-a, OPN-b, and OPN-c splice variants [16,18,19]. More recently, OPN-SV expression has also been reported in mouse metastatic and primary breast tumors [20].

The expression of OPN-SV has also been evaluated in some human cardiac pathological conditions. OPN-a, OPN-b and OPN-c are differentially expressed during calcific aortic valve disease (CAVD) progression and can inhibit biomineralization [21]. OPN-SV expression has also been studied in heart failure of different origins. OPN-a is expressed in higher levels in dilated compared to ischemic cardiomyopathies. Otherwise, OPN-b and OPN-c were only detected in ischemic cardiomyopathy. These data evidenced the correlation of the expression of these OPN-SV with different cardiac clinical-pathophysiological settings [22]. Human OPN-SV also exerts divergent effects on neovascularization through differential effects on arteriogenesis and macrophage accumulation in vivo and on macrophage migration and survival. In this experimental model, OPN-a and, most importantly, OPN-c significantly increased macrophage migration [23]. Moreover, OPN-a stimulates pro-inflammatory signaling in both macrophages and myoblasts, possibly through induction of TNC–TLR4 signaling, opening possibilities to target this splice variant and related pathways in the damaged muscle microenvironment as a therapeutic approach [24].

Alternative translation from the different start codons of the single OPN messenger RNA (mRNA) can generate two OPN isoforms, a secreted OPN (sOPN) and an intracellular OPN (iOPN) (Figure 2), through a molecular mechanism elegantly reported by Shinohara et al. (2008) [25]. The sOPN is the longest isoform, containing the N-terminal SP sequence corresponding to exon 2, allowing targeting to secretory vesicles and cytokine secretion. Otherwise, iOPN is the shortest one, lacking the sOPN N-terminal SP, as a result of a non-AUG alternative translation from a downstream start site sequence, and is mainly localized into the cytoplasm [25,26].

**Figure 2 ijms-24-02942-f002:**
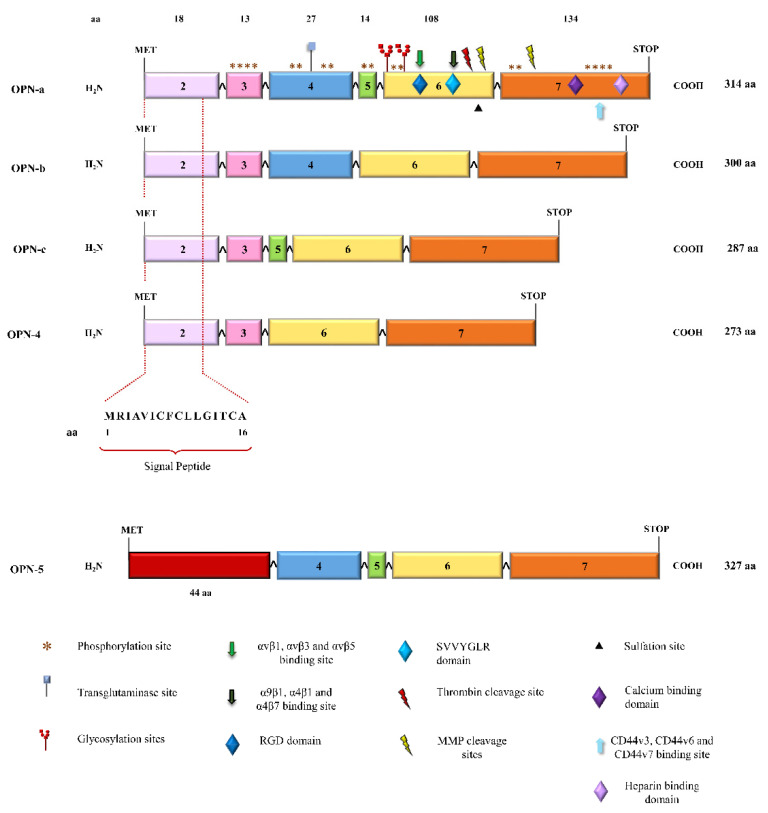
Schematic representation of OPN splice variants. The canonical translation start site is located on exon 2, with the first 16 amino acids coding for the signal peptide (SP). Exon 4 has an important site for transglutaminase binding. Exon 6 has glycosylation, sulfation, thrombin, and MMP cleavage sites. Exon 6 has the RGD and the SVVYGLR domains, regions where protein binds to different integrins. Exon 7 has calcium and heparin-binding domains and a site for MMP cleavage. Phosphorylation regions are scattered throughout the protein, and each * means one phosphorylation site. This Figure was partially based on information provided in previous reports [18,27,28].

Considering the existence of both sOPN and iOPN, we recently analyzed OPN-SV amino acid sequences obtained from NCBI using some software that predicts SP sequences and the location of their cleavage sites, as well as their putative subcellular localization. The collection of these data is shown in Table 1.

We found that among all reported OPN-SV annotated amino acid sequences, only OPN-5 does not contain the SP. The OPN-4, OPN-a, OPN-b, and OPN-c splice variants contain SP eukaryotic consensus sequences, besides also displaying peptidase cleavage sites at the same amino acid position, in between amino acids 16 and 17 of the SP sequence. Further evaluating these amino acid sequences to predict OPN putative subcellular localization, we found that although OPN-5 does not contain a predicted SP sequence, it has been predicted that it displays an 81% probability of being secreted. Otherwise, the remaining isoforms have higher scores rates for being secreted. Moreover, these 5 OPN-SV also contain predicted nuclear localization sequences (NLS) and calculated scores that indicate it is partially localized to the nucleus, or it can be localized to both the nucleus and the cytoplasm and even into the mitochondria. It has also been recently described that OPN-5 and its corresponding sub-variants (OPN-5b, OPN-5c, OPN-5d, and OPN-5e) are also secreted [12]. We then hypothesize that most OPN-SV protein sequences could generate both secreted and intracellular isoforms after SP proteolytic cleavage.

Although most studies refer to sOPN, the analysis of OPN total transcripts and protein expression do not distinguish between the sOPN isoforms that are still inside the cells and will be secreted later and the iOPN isoforms. Hence, for didactical purposes, we herein define as the sOPN the data of variants that relate to the isoforms that can extracellularly bind to cell receptors and/or are present in the conditioned media (CM) secreted by cells. Otherwise, those OPN isoforms detected inside the cells (even if they will be secreted later), such as in the cytoplasm, in the nucleus, or any other cellular compartment, were considered herein as iOPN.

## 2. Secreted Roles of OPN

Most of OPN’s roles at the cellular level have been attributed to the extracellular effects that occur after secretion, where receptor binding results in the activation of specific intracellular signaling pathways. Indeed, OPN receptors include α_v_ (β_1_, β_3_, or β_5_) and (α_4_, α_5_, α_8_, or α_9_) β_1_-integrins, the variants 6 and 7 of the hyaluronan receptor CD44, and also the epidermal growth factor receptor (EGFR) [34]. Until the first description of iOPN, the studies regarding OPN were focused on the cytokine-like properties of this protein [35,36]. Different signaling pathways, such as NF-κB, PKB/Akt, and MAPK, mediate cancer-related aspects, such as angiogenesis, invasion, and metastasis. In this context, some molecules, such as soy isoflavone genistein, have been reported to regulate these pathways by controlling the expression of secreted OPN expression in breast cancer metastatic cells [37].

In this section, we only considered the studies that assessed OPN proteins in the extracellular microenvironment, either using exogenous addition of OPN (recombinant OPN or using a conditioned medium) and evaluating the effects of OPN by binding to its receptors.

### 2.1. Secreted OPN Activities Mediated by the Integrin Receptors

Amongst the wide variety of integrins by which OPN and OPN-SVs can bind to exert their effects on different cancer models, αvβ3 is the most frequently found. In a melanoma cancer model, for instance, by binding to αvβ3, OPN induces MEKK1-dependent JNK1 phosphorylation [38]. In gliomas, OPN directly stimulates angiogenesis via the αvβ3/PI3K/AKT/eNOS/NO signaling pathway and may play an important role in tumorigenesis by enhancing angiogenesis in gliomas [39,40] In tumor angiogenesis, malignancy, invasion, and metastasis; hypoxia is a unique parameter. The key regulator of hypoxia phenomena is hypoxia-inducible factor 1-α (HIF1α). Under hypoxia, OPN regulates HIF1α-dependent VEGF expression via inducing α_v_β_3_-linked kinase (ILK)/Akt-mediated nuclear factor (NF)-κB p65 activation [41].

Taking into consideration data regarding secreted OPN-SV and its effects, the full-length OPN-a splice variant acts as an oncogene in lung cancer cells. Higher levels of OPN-a are associated with poor clinical outcomes in these patients, and αvβ3 integrin is involved in the OPN-a-mediated malignant behavior of these tumor cells. The interaction of OPN and αvβ3 integrin has a role in lung cancer cell adhesion to the bone matrix, which is probably one of the early steps of bone metastasis formation in lung cancer [42].

Other authors reported that in esophageal adenocarcinoma, the OPN-b and OPN-c splice variants have distinct pro-invasion and dissemination phenotypes. In vitro and in vivo ectopic expression of OPN-b significantly enhanced cell migration and adhesion to laminin, while OPN-c overexpressing cells showed decreased cell migration and increased cell detachment. Moreover, by inhibiting the RGD motif, cell migration, adhesion, and clonogenic survival promoted by OPN-b were significantly abrogated, but not the OPN-c mediated phenotypes. These data evidenced that OPN-b, but not OPNc, acts through integrin-dependent signaling in esophageal adenocarcinoma [43]. High expression levels of *OPN-a*, *OPN-b*, and especially *OPN-c*, as opposed to low expression of *OPN-4* and *ITGA2* integrin variants, are associated with an advanced stage of tumor progression and poor prognosis in melanoma [44].

By binding to integrin α9β1, OPN increases angiogenesis and motility of melanoma cells through the secretion of COX-2 and PGE2, activating ERK and p38-dependent AP-1 [45]. Integrin receptors also mediate the roles of sOPN in immune regulation. Secreted OPN binds to monocytes, but not resting T cells, NK cells, or B cells and mediates chemoattraction of IL-1-activated human monocytes. Schack L and co-workers [46]. identified an αXβ2 integrin (CD11c/CD18), which is highly expressed on the cell surface of monocytes, as a novel OPN receptor that is required for OPN-mediated phagocytosis, thereby elucidating an important mechanism of an innate immune function of OPN [46]. In neutrophils, OPN has chemotactic activity. This effect can be explained by extracellular post-translational modification, polymerization, and/or thrombin cleavage. In these immune cells, OPN polymerizes in vivo and plays a critical role in the chemotactic activity. In leukocytes, OPN has this polymerization role dependent on α9β1 integrin [38]. Secreted OPN binding to integrin receptors on macrophages is also a hallmark of OPN roles in these cells. For instance, OPN treated-bone marrow-derived macrophages (BMDMs) were challenged with growth factor withdrawal and neutralizing integrin antibodies. Authors have found that the survival of BMDMs is mediated primarily through the α4 integrin. Moreover, in chemotaxis studies, migration to OPN was blocked by neutralizing α4 and α9 integrin antibodies. Further, OPN did not affect macrophage activation, as measured by IL-12 production. Additionally, the relative contributions of the RGD and the SLAYGLR functional domains of OPN to leukocyte recruitment were evaluated in an in vivo model. By generating chimeric mice expressing mutated forms of OPN in myeloid-derived leukocytes, and found that the SLAYGLR functional domain of OPN, but not the RGD, mediated macrophage accumulation in response to thioglycollate-elicited peritonitis. Taken together, these data indicated that OPN interacts with α4 and α9 integrins via the SLAYGLR domain playing a key role in macrophage biology by regulating migration, survival, and accumulation [40].

Secreted OPN also plays a role in rat mesenchymal stem cells (rMSCs) inducing migration through β1 integrin binding. In this model, OPN promotes rMSCs migration by activating FAK and ERK pathways, which may be attributed to changes in cell stiffness caused by the reduction in the amount of organized actin cytoskeleton [47].

### 2.2. Secreted OPN Activities Mediated by the CD44 Receptor

CD44 is one of the best-characterized receptors by which OPN acts on tumor progression. Studies have been reporting that OPN upregulation induces CD44 expression, which in turn increases OPN expression in an autocrine manner [48,49]. OPN binds to CD44, leading to many different effects in cells, including invasion, focus formation, proliferation, invasion, motility, and clonogenicity. In a cancer progression model using NIH3T3 fibroblastic cells containing Ras family mutants (H-RasV12 and Rit79L), for instance, OPN is overexpressed, promoting overexpression of CD44. By employing an antibody to inhibit CD44 or short interfering RNA from knocking down OPN expression, cell invasion and foci formation induced by H-RasV12 were reduced, requiring an intact OPN-CD44 system [48]. In thyroid carcinoma, OPN also has a functional role in the OPN/CD44 axis. It has been shown that rearrangement of the (RET/PTC) signaling triggers the formation of an autocrine loop involving OPN and CD44 that stimulates the proliferation and motility of PCCl3 rat cells [49]. These same authors also reported that the treatment of human papillary thyroid cancer cells with recombinant exogenous OPN stimulated matrigel invasion and activated the ERK and AKT1 signaling pathways, while the blockage of anti-CD44 antibodies prevented these effects [50]. In a hepatocellular carcinoma (HCC) model, OPN from the HCC cell line conditioned media (CM) induced cell proliferation in a paracrine manner. This effect was demonstrated to be dependent on CD44, once only CD44 positive cell lines responded to CM rich in OPN, while CD44 knockdown blocked the proliferative effects [51].

Through binding to the CD44 receptor, OPN activated c-jun-NH2-kinase signaling and promoted the clonogenicity of colorectal cancer (CRC) cells. In this study, macrophages, when co-cultured with CD44-positive CRC cells, were able to produce higher levels of OPN, which in turn facilitated the tumorigenicity and clonogenicity of these cells. The knockdown of CD44 or treatment with CD44-blocking antibodies attenuated OPN secretion, further reinforcing that CD44 can modulate OPN expression [52]. Human recombinant OPN (rhOPN) also promoted cell proliferation, migration, and invasion, accompanied by upregulation of ALDH1- positive cancer stem cells in CRC cells through the activation of PI3K/Akt/GSK/3β-β-catenin pathway [53].

### 2.3. Secreted OPN Activities and Transactivation of EGFR

Transactivation of the epidermal growth factor receptor (EGFR) by OPN is another mechanism by which OPN plays its role. Lee S. J. and co-workers [54] reported that OPN has an important role in vascular smooth muscle cells (VSMC) proliferation induced by 4-hydroxynonenal, which is mediated by activation of AP-1 and C/EBPβ through EGFR pathways. An in vitro study further speculated that OPN might also bind to integrin β1 to maintain the activation of EGFR, which allows the proliferation of the human prostate tumor cells [55]. Further reinforcing OPN and EGFR transactivation, some reports have been demonstrating a significant correlation between the expression of these markers in some cancer types, such as in hepatocellular carcinoma and clear cell renal cell carcinoma [56,57].

### 2.4. Secreted OPN Activities Mediated by More Than One Receptor

In some specific experimental models, OPN can also exert its functions by binding to more than one receptor, individually or at the same time. In PC3 human prostate cancer cells, for instance, OPN can activate Akt and Raf/MEK/ERK pathways either through the αvβ3 integrin or the CD44 cell surface receptor, leading to enhanced proliferation and survival (anti-apoptotic) effects [58]. A further example in cancer cells is mediated by the human recombinant OPN, which induces MMP-2 upregulation through the SDF-1a/CXCR4 axis, and by binding to integrin αvβ3 and CD44v6. In this way, it activates the PI3K/Akt and JNK pathways in HepG2 and SMMC7721 cells, enhancing HCC cell invasion [59]. The combined action of OPN-mediated receptors has also been reported in an osteoclast cell model, in which OPN can induce the expression of its receptors in an autocrine manner. The exogenous addition of OPN to OPN-deficient osteoclasts increased the surface expression of CD44 and rescued osteoclast motility due to the activation of αvβ3 integrin [60]. OPN roles in cancer progression can also be mediated by the integration of multiple signals, in which OPN could act as a scaffold on multiple cell receptors, including ICOSL, CD44, and several integrins. COSL/ICOS are costimulatory molecules pertaining to immune checkpoints, and their binding transduces signals having anti-tumor activity. In a mouse breast cancer metastatic model, for instance, the ICOSL is a receptor for OPN, interacting at a different domain than that used by ICOS. ICOSL binding by ICOS or OPN exerts opposing effects on breast cancer cell migration, which is induced by OPN and dominantly inhibited by ICOS [61].

## 3. Intracellular OPN and Diseases

The iOPN was first described by Zohar et al. (1997) [62], and current data regarding iOPN are summarized in Table 2. By evaluating the possible relationships between OPN expression and stages of osteogenic cell differentiation in cell populations derived from fetal rat calvarial cells, it was found that iOPN displayed a perinuclear and a perimembranous distribution, in which patches of OPN were concentrated at the cytoplasmic cell surface. It also demonstrated the co-localization of iOPN, CD44, and the complex containing the actin-binding proteins ezrin, radixin, and moesin in migrating fibroblasts [63]. The co-localized iOPN and CD44 were also observed in activated macrophages, migrating osteoclasts, and metastatic cells, which was consistent with the induction of CD44 and iOPN expression and the migratory properties of these cells [63,64]. Moreover, the iOPN can also modulate CD44-dependent chemotaxis in mice peritoneal macrophages [64]. Intracellular OPN has also been associated with the transformation of chondrocytes into cells presenting osteocytic phenotype [65].

Human iOPN was also found to be expressed in human cell lines and tissues of oral squamous cell carcinoma (OSCC) and seems to be associated with tumorigenesis in this cancer type once not detected at the normal oral mucosa or gingiva [66]. In colorectal carcinoma (CCR), co-expression of cytoplasmic OPN and nuclear β-catenin has been associated with unfavorable prognostic factors [67]. In central nervous system (CNS) lymphoma, the balance between sOPN and iOPN activities has a pro-tumorigenic role, in which iOPN causes transcriptional downregulation of the NF-κB inhibitors, A20/TNFAIP3 and ABIN1/TNIP1, while sOPN promotes receptor-mediated activation of NF-κB [68]. A similar combined action of sOPN and iOPN is associated with head and neck (HNC) cancer progression induced by cancer-associated fibroblast-derived IL-6. The IL-6 induced-sOPN could promote HNC progression via the integrin αvβ3-NF-κB pathway, and the combination of iOPN and IL-6 cytoplasmic evaluation had a better prognostic and diagnostic performance in HNC than either molecule alone [69]. The concerted action of sOPN and iOPN similarly control hepatitis C virus (HCV) replication and assembly in hepatocellular carcinoma (HCC)-infected cells. In response to interactions of sOPN with the cell surface receptors αvβ3 and CD44, the iOPN isoform associates with some HCV proteins (including NS5A and core proteins) in the endoplasmic reticulum and lipid droplets, the specialized sites where HCV replication and assembly occur [70]. In this same tumor type, iOPN may function as a negative regulator of Toll-like receptor (TLR)-mediated immune responses to ameliorate inflammation-associated cytokines, favoring mice hepatocarcinogenesis [71].

**Table 2 ijms-24-02942-t002:** Main current findings regarding iOPN and associated roles.

iOPN Expression Pattern/Role	Main Findings	Cell Type	Approach/Methodology	Reference
OPN displayed a perinuclear and a perimembraneous distribution	iOPN was associated with stages of osteogenic cell differentiation	Fetal rat calvarial cells	Single-cell analysis, flow cytometry, and confocal microscopy	[62]
iOPN localizes in the cytoplasm but outside secretory vesicles	iOPN is translated from a non-AUG codon, accompanied by deletion of the N-terminal 16-aa signal sequence.	Dendritic and T cells	R’ RLM-RACE, in vitro translation, confocal microscopy, *Ifna4* promoter reporter assay, isolation of secretory vesicles; in vivo assays, ELISA and intracellular cytokine flow cytometry, immunoblotting, RT-qPCR	[25,26]
Soluble OPN altered the actin cytoskeleton of tumor cells	iOPN induces rapid Tyr-418 dephosphorylation of c-Src, with decreases in actin stress fibers and increased binding to the vascular endothelium	Human melanoma and sarcoma cell lines; and patient-derived samples	Functional assays in vitro, in tumor cells, in mouse models, and ex vivo, using exogenous OPN or negative controls, including a site-directed mutant OPN	[72]
iOPN co-localizes with fungal PRRs in macrophages stimulated by *Pneumocystis*	iOPN is essential for generating antifungal innate immune responses in PRR recognition, signal transduction, phagocytosis, clearance of *Pneumocystis*, and cytokine production	Cells obtained from C57BL/6, *Opn*^−/−^, *Rag2*^−/−^, and *Opn^−/−^ Rag2^−/−^* mice	RT-qPCR, confocal microscopy, immunoprecipitation, immunoblotting, ELISA, and analyses of phagocytosis, ROS production, and *Pneumocystis* clearance	[73]
iOPN expression is enhanced by VSV and SeV virus infection	iOPN acts as a positive regulator in innate antiviral immunity through the stabilization of TRAF3		iOPN ectopic overexpression or OPN deficiency or knockdown; SPP1 knockout mice; Immunoprecipitation and immunoblotting; ELISA; RT-qPCR; ubiquitination assays	[74]

OPN: osteopontin; iOPN: intracellular OPN; pattern recognition receptor (PRR); plasmacytoid dendritic cells (pDC); Sendai virus (SeV); vesicular stomatitis virus (VSV); tumor necrosis factor receptor (TNFR)-associated factor 3 (TRAF3); Cytoplasmic OPN has also been reported to be differentially expressed at the pyramidal neurons of the hippocampus of Alzheimer’s disease (AD) compared with age-matched control brains. Moreover, cytoplasmic OPN staining intensity was significantly associated with amyloid-β load and aging among all control and AD subjects [75]. It has also been shown that exhaustion of cytoplasmic OPN expression is associated with the loss of its protective role in the pancreas islets cells and with destructive insulitis and diabetes [76].

Nuclear localization of iOPN has also been observed. In embryonic kidney 293 cells, OPN accumulates in the nuclei and is correlated with chromatin condensation during cell division, exhibiting a subcellular distribution that coincided with polo-like kinase-1 (Plk-1), an established regulator of centrosome-related events and mitosis [77]. Moreover, OPN splice variants detected inside the cells differently contribute as breast cancer prognostic markers. The nuclear OPN-c isoform is strongly associated with mortality in patients with early breast cancer, while cytoplasmic OPN-a and OPN-b predict poor outcomes. By contrast, iOPN does not correlate with prognosis [78]. A similar result was demonstrated by other authors, in which most CRC samples showed high OPN nuclear expression, but it did not mean any prognostic value. However, cytoplasmic OPN overexpression was associated with a favorable prognostic in this tumor model [79]. In ovarian cancer, high nuclear OPN-c staining patterns were observed in patients that responded to platinum or pegylated liposomal doxorubicin (PLD) treatment, and this pattern was associated with longer overall survival and progression-free survival (PFS). It was also found that longer PFS was also associated with high expression of both nuclear and cytoplasm OPN-c, both in platinum-resistant and in those patients that responded to PLD [80].

It has also been reported that nuclear localization of OPN in cardiomyocytes is a marker of severe heart failure and cardiac allograft vasculopathy [81].

## 4. Intracellular OPN and Immunity

Several reports also described the roles of iOPN in the immune system and many inflammatory and immune-modulating disorders. These studies demonstrated that innate immune phagocytes, such as macrophages and dendritic cells (DCs), constitutively express high levels of both iOPN and sOPN, while T cells preferentially express sOPN. Shinohara and co-workers [82] reported that iOPN selectively coupled TLR9 signaling to express IFN-α in plasmacytoid dendritic cells (pDC). Moreover, the co-localization of iOPN and MyD88-containing signal transduction complex after TLR9-iOPN mediated engagement was essential for efficient nuclear translocation of the transcription factor IRF7 and associated IFN-α gene expression in these cells [82]. The inhibition of the TLR9-MYD88-STAT3 signaling pathway is also associated with the role of B cell iOPN in protecting from autoimmunity-driven mouse lymphoma development [83]. Inhibition of iOPN expression mediated by Type-I IFN receptor (IFNAR) expressed by DCs de-repressed interleukin-27 (IL-27) secretion, and as a result, prevented Th17 responses in vitro and in vivo, which, when aberrantly expressed, has been associated to the generation of autoimmune diseases [25]. Further, these same authors demonstrated that the coordinated action of sOPN and iOPN regulates the functional phenotype of DCs subsets. The shortened OPN translation isoform activates *Ifna4* gene expression and podosome formation in pDC, contributing to the biological activities of this DC subset. These findings indicate that factors that alter the translational balance of OPN in favor of either iOPN or sOPN may contribute to the phenotype of activated DCs [25].

Another role of iOPN in the immune system was described by Leavenworth et al. (2014) [84]. NK cells display an important role in the immune system’s response against viruses and cancer. In this report, the authors demonstrated the potential contribution of iOPN in NK cell homeostasis, survival, and development through IL-15 ligation. NK cells overexpress iOPN, and in the absence of this protein, NK cells exhibit a decrease in the ability to develop memory-like cells and respond to viral infection and metastatic tumors. Moreover, the iOPN deficient phenotype increases NK cell death and impaired NK cell expansion and differentiation.

In the context of follicular helper T (TFH) and follicular regulatory T (TFR) cells, responsible for regulating humoral immunity, it was demonstrated that iOPN and its association with p85α are crucial to the differentiation of these TF cells [85]. Bcl-6, a transcription factor that regulates TF cell differentiation, is protected by iOPN from ubiquitin-dependent proteasome degradation.

The imbalance between myeloid and lymphoid cells directly impacts immune responses; thus, hematopoiesis is tightly controlled. Inflammations, infections, and irradiation require a greater immune system response through increased activation of myelopoiesis to compensate for the loss of myeloid cells. However, lymphocytes have distinct mechanisms from myeloid cells, and a normal immune system keeps an optimal balance between myeloid and T cells through the activation of some proteins, such as OPN. Other authors found that OPN regulates cell population balance through an increase in lymphoid and a decrease in myeloid cells. During lymphoid cell expansion, irradiation, and systemic fungal infection, iOPN induces a negative regulation of myelopoiesis, while sOPN stimulates lymphoid cell expansion. Both OPN forms work together in this balance despite having different roles [86].

OPN also has a protective role in infectious diseases, including iOPN involvement in the host response against fungi. iOPN is essential for generating antifungal innate immune responses in fungal pattern recognition receptors (PRR) (such as dectin-1, Toll-like receptor 2 (TLR2), and mannose receptor (MR) recognition and clustering on the cell surface, signal transduction, phagocytosis, fungi clearance and cytokine production [73,87].

Furthermore, it was demonstrated that iOPN is an essential positive regulator in innate antiviral immunity by activating IFN-β through a new regulatory mechanism in innate antiviral signaling through iOPN-mediated stabilization of tumor necrosis factor receptor (TNFR)-associated factor 3 (TRAF3). In this study, the knockdown of OPN significantly attenuated virus-induced IFN-β production and enhanced vesicular stomatitis virus (VSV) replication, while overexpression of iOPN showed the opposite effect [74].

## 5. Conclusions

OPN has been classically classified as an extracellular cytokine. However, as a result of several post-transcriptional and post-translational mechanisms, different OPN variants can be formed, including the sOPN and iOPN isoforms. Amongst these variants, the OPN-SVs have been extensively characterized in distinct tumor models and display tissue and tumor-specific roles. By analyzing OPN-SV amino acid sequences as well as their expression patterns, current data provide evidence that most of them display a high probability of being secreted, although OPN has also been detected in distinct cellular compartments, mainly perimembrane, cytoplasmic, and nuclear. The iOPN isoforms have been identified in distinct cellular contexts, such as modulating tumorigenesis, controlling viral infection and assembly, regulating immune responses, neurological disorders, infectious diseases, and gene expression. In association with signaling pathways induced by sOPN and their corresponding receptor binding, both sOPN and iOPN have several key roles in some experimental models. These expression patterns and mechanisms, in addition to sOPN and their variant’s specific localization, are summarized in Figure 3. Hence, a better understanding of the iOPN roles and their association with sOPN may provide additional evidence for using these molecules as disease markers and putative targets for new therapeutic approaches that could specifically target sOPN or iOPN.

## Figures and Tables

**Figure 3 ijms-24-02942-f003:**
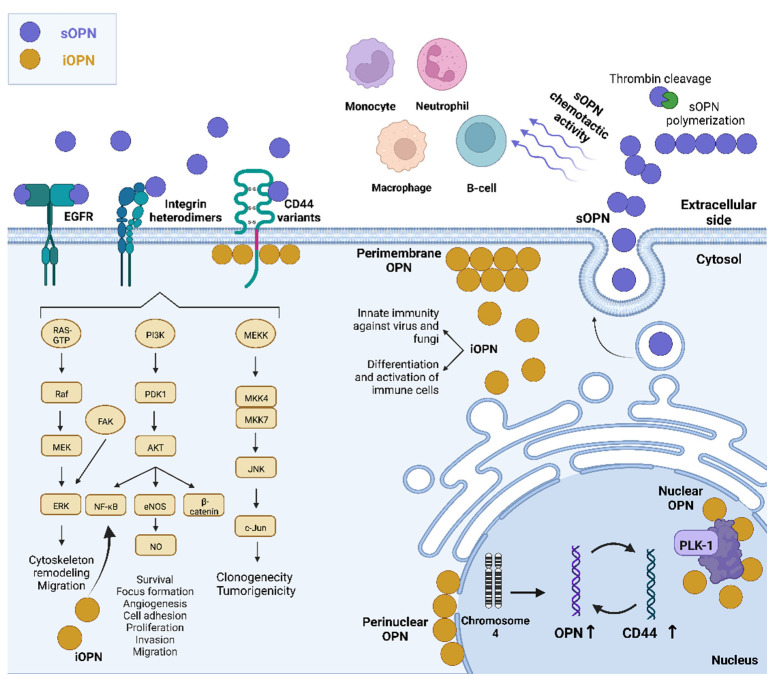
Roles and mechanisms of osteopontin in cancer cells. Osteopontin is a highly expressed protein that acts through several mechanisms in cancer cells. Secreted OPN can regulate PI3K/AKT, MAPK, and JNK intracellular signaling pathways through interacting with its known membrane receptors, CD44 variants, integrins, and/or EGFR, resulting in the activation of a pro-tumorigenic phenotype, including increased survival, proliferation, clonogenicity and capacity of migration and invasion. Extracellular OPN has been extensively associated with the immune system once this protein mediates chemoattraction of cells, such as monocytes, neutrophils, and lymphocytes, as well as macrophage accumulation in diverse diseases, including in tumor microenvironments. This variety of extracellular roles is due to several modifications, such as post-translational mechanisms, polymerization, and thrombin cleavage. Intracellular OPN has been localized in the perimembranous and perinuclear regions regulating the expression of NF-kB, cytoskeleton remodeling through the CD44 receptor, and activation and differentiation of immune cells in responses against viruses and fungi. Besides cytoplasmic OPN, it has been observed a nuclear localization for this protein where it accumulates and is associated with the presence of Plk-1 and chromatin condensation. Once OPN has been associated with a poor prognosis and acquisition of pro-tumorigenic features, molecules aiming the regulation of OPN have been the focus of studies, such as genistein, which inhibits sOPN expression, leading to reduces colony formation rates, migration, and invasion. sOPN: secreted OPN. iOPN: intracellular OPN. Created with BioRender.com (accessed on 11 November 2022).

**Table 1 ijms-24-02942-t001:** Predictions for SP and subcellular localization for OPN-SV protein sequences.

OPN-SV	N-Terminal Sequence	Signal-6.0 SP Eukaryotic Consensus Sequence	Signal-6.0 Peptidase Cleavage Site 16/17	PrediSi Probability of SP Prediction	DeepLoc Probability of Being Secreted	MULocDeep Probability of Subcellular Localization	NucPred Probability of Nuclear Localization
OPN-a	MRIAIVICFCLLGITCAI	yes	yes	84.0%	>90%	100% secreted	61%
OPN-b	MRIAIVICFCLLGITCAI	yes	yes	84.0%	>90%	100% secreted	60%
OPN-c	MRIAIVICFCLLGITCAI	yes	yes	84.0%	>90%	100% secreted	54%
OPN-4	MRIAIVICFCLLGITCAI	yes	yes	84.0%	>90%	100% secreted	55%
OPN-5	MGIVPRSLDKKAHRVQFQ	No	No	0.0%	81%	<10% secreted; ~30% mitochondrial; ~20% nuclear	65%

The OPN-SV amino acid sequences were analyzed by software that predicted the SP sequences (SignalP 6.0 [29]) and corresponding peptidase cleavage sites (PrediSi [30]). SignalP 6.0 also identifies the position of the amino acid sequence in which is located the peptidase cleavage site (in between amino acids 16 and 17 of the OPN SP sequence). Other software predicted the probability of each OPN-SV to be secreted (DeepLoc [31]), their subcellular (MULocDeep [32], or nuclear localization (NucPred [33]). The N-terminal end of each OPN-SV amino acid sequence is shown; SP: signal peptide; Yes: presence of SP; No: absence of SP; >: probability higher than; The NCBI reference number for each amino acid sequence analyzed are as follows: OPN-a, NP_001035147.1; OPN-b, NP_000573.1; OPN-c, NP_001035149.1; OPN-4, NP_001238758.1, and OPN-5, NP_001238759.1.

## Data Availability

Not applicable.

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
