# Peer review of "The Intracellular and Secreted Sides of Osteopontin and Their Putative Physiopathological Roles"

_ijms, 2023, doi:10.3390/ijms24032942_

Round 1
Reviewer 1 Report
I had the pleasure to review the manuscript “The intracellular and secreted sides of osteopontin”. The purpose of the review was to generate a manuscript that describes the role, mechanistic pathways and associations of osteopontin (OPN) in oncology and immunology. The presentation of the evidence turned around the concept of intracellular and secreted OPN.
I consider that the authors did a good job describing the biology of the OPN molecule from the potential transcriptional events to the receptors’ interactions, and cascade of intracellular events. I just have a couple of comments:
Figures 1, 2 and 3 are of extremely low quality. Hard to read and understand. Although the information in there seems to be OK, they are too low quality for a publication. I will need to review them again after the resolution is improved.
I suggest the possibility to add more information to the analysis in table 1. Since the probability of the OPN to be secreted vs intracellular localization is the main topic of your paper. Can you add information of the analysis of this public data by organ/cell type? Or potentially tumor type? I would expect that some of the biological of this protein can be either organ or tumor-specific.
Author Response
Responses to Reviewer 1 Comments
# Point 1:
“English language and style are fine/minor spell check required”
Answer to # Point 1: Thanks to re reviewers for this comment. Accordingly, we checked spelling and grammar, and all changes are highlighted at the revised manuscript.
# Point 2:
“Figures 1, 2 and 3 are of extremely low quality. Hard to read and understand. Although the information in there seems to be OK, they are too low quality for a publication. I will need to review them again after the resolution is improved.”
Answer to # Point 2: We apologize for this, but we do not understand why the images were extremely low quality in our submitted version. Perhaps because we had to attach de word version instead of a pdf version. For us, at the word version image quality is nice. To clarify image analysis and quality, we attached the three original high-quality images. Also, we provided you a pdf version of the revised manuscript, hoping it will improve the images quality in this revised version of our manuscript.
# Point 3:
“I suggest the possibility to add more information to the analysis in table 1. Since the probability of the OPN to be secreted vs intracellular localization is the main topic of your paper. Can you add information of the analysis of this public data by organ/cell type? Or potentially tumor type? I would expect that some of the biological of this protein can be either organ or tumor-specific.”
Answer to # Point 3: We kindly appreciate this reviewer’ comments, but we clarify that we analyzed the sequences were obtained from NCBI, with the reference numbers: OPN-a, NP_001035147.1; OPN-b, NP_000573.1; OPN-c, NP_001035149.1; OPN-4, NP_001238758.1, and OPN-5, NP_001238759.1. Perhaps we did not use the correct term at the submitted manuscript, when wriiting “public database”, which actually were the NCBI mentioned sequences. We agree with the reviewer suggestion that it is possible that these amino acid sequences may display tumor and/or organ-specific localization. However, using the NCBI sequences we analyzed, we cannot conclude about this hypothesis. Our aim with this analysis and Table 1 was to provide some evidence regarding secreted or intracellular localization based on protein SP consensus signal sequences and their corresponding cleavage sites, and based on this, provide some evidence regarding their putative localization.

Reviewer 2 Report
The manuscript is well-written and highlights significant research in the field. However, most of the studies are associated with cancer and immune responses with minimal studies on other diseases. I recommend changing the title of the manuscript to reflect the same.
On page 8, the authors talk about using software to analyze the public database. The software details are provided in the table but the authors are requested to provide the details on what database was used for analysis.
Most of the cited work is old with approximately 20 of the cited 89 references reported in the last 5 years. This again falls in alignment with the first comment about the manuscript’s focus. If the authors are trying to keep the subject broader, please consider adding newer references. Some of the newer works are provided here for reference,
a) Rizzello, Celeste, et al. "Intracellular osteopontin protects from autoimmunity-driven lymphoma development inhibiting TLR9-MYD88-STAT3 signaling." Molecular Cancer 21.1 (2022): 1-21.
b) Sulsenti, Roberta, et al. "The protective role of mast cells against neuroendocrine prostate cancer depends on the release of cytokines mediated by intracellular osteopontin." Cancer Research 82.12_Supplement (2022): 2556-2556.
Here is another review that focuses explicitly on the role of osteopontin in cancer - Kariya, Yoshinobu, and Yukiko Kariya. "Osteopontin in Cancer: Mechanisms and Therapeutic Targets." International Journal of Translational Medicine 2.3 (2022): 419-447.
Please provide the relevant references for the following lines-
a) Page 2, Introduction, 3rd paragraph, line 2-3.
b) Page 2, Introduction, 4th paragraph, line 1.
All figures are pixelated making it hard to visualize and read. Please provide a sharper image or redo the image with a higher resolution. Additionally, since these images were remade from cited literature, provide relevant references in the figure legend.
Author Response
Responses to Reviewer 2:
# Point 1:
“English language and style are fine/minor spell check required”
Answer to # Point 1: We also kindly thank the revisor for this point. As answered to reviewer 1, we checked spelling and grammar, and all changes are highlighted at the revised manuscript.
# Point 2:
“The manuscript is well-written and highlights significant research in the field. However, most of the studies are associated with cancer and immune responses with minimal studies on other diseases. I recommend changing the title of the manuscript to reflect the same. “
Answer to # Point 2: We appreciated this point and accordingly we changed the title to “The intracellular and secreted sides of osteopontin and their putative physiopathological roles”. We hope this fits we the reviewer expectations and suggestions.
# Point 3:
“On page 8, the authors talk about using software to analyze the public database. The software details are provided in the table but the authors are requested to provide the details on what database was used for analysis.”
Answer to # Point 3: We again thank the reviewer for this additional point. As answered at # Point 3 to the reviewer 1, we actually used the sequences from NCBI, which we named “public database”. To better clarify this, we revised the text and used “Considering the existence of both sOPN and iOPN, we recently analyzed OPN-SV amino acid sequences obtained from NCBI database using some software that predicts SP sequences.”
We copy here the answer we provided to reviewer 1 regarding this point: “We kindly appreciate this reviewer’ comments, but we clarify that we analyzed the sequences were obtained from NCBI, with the reference numbers: OPN-a, NP_001035147.1; OPN-b, NP_000573.1; OPN-c, NP_001035149.1; OPN-4, NP_001238758.1, and OPN-5, NP_001238759.1. Perhaps we did not use the correct term at the submitted manuscript, when wriiting “public database”, which actually were the NCBI mentioned sequences. We agree with the reviewer suggestion that it is possible that these amino acid sequences may display tumor and/or organ-specific localization. However, using the NCBI sequences we analyzed, we cannot conclude about this hypothesis. Our aim with this analysis and Table 1 was to provide some evidence regarding secreted or intracellular localization based on protein SP consensus signal sequences and their corresponding cleavage sites, and based on this, provide some evidence regarding their putative localization.”
# Point 4:
“Most of the cited work is old with approximately 20 of the cited 89 references reported in the last 5 years. This again falls in alignment with the first comment about the manuscript’s focus. If the authors are trying to keep the subject broader, please consider adding newer references. Some of the newer works are provided here for reference,
- a) Rizzello, Celeste, et al. "Intracellular osteopontin protects from autoimmunity-driven lymphoma development inhibiting TLR9-MYD88-STAT3 signaling." Molecular Cancer 21.1 (2022): 1-21.
- b) Sulsenti, Roberta, et al. "The protective role of mast cells against neuroendocrine prostate cancer depends on the release of cytokines mediated by intracellular osteopontin." Cancer Research 82.12_Supplement (2022): 2556-2556.
Here is another review that focuses explicitly on the role of osteopontin in cancer - Kariya, Yoshinobu, and Yukiko Kariya. "Osteopontin in Cancer: Mechanisms and Therapeutic Targets." International Journal of Translational Medicine 2.3 (2022): 419-447.
Please provide the relevant references for the following lines- a) Page 2, Introduction, 3rd paragraph, line 2-3.
b) Page 2, Introduction, 4th paragraph, line 1.”
Answer to # Point 4: We kindly thank the reviewer for this point and accordingly we updated some references, which are highlighted throughout the manuscript.
Regarding the reference “Sulsenti, Roberta, et al. "The protective role of mast cells against neuroendocrine prostate cancer depends on the release of cytokines mediated by intracellular osteopontin." Cancer Research 82.12_Supplement (2022): 2556-2556”, we found at the link https://aacrjournals.org/cancerres/article/82/12_Supplement/2556/704260 it is an abstract and not a full paper. Because of this, we opted not to cite this work at this moment.
# Point 5:
All figures are pixelated making it hard to visualize and read. Please provide a sharper image or redo the image with a higher resolution. Additionally, since these images were remade from cited literature, provide relevant references in the figure legend.
Answer to # Point 5: Thanks for this additional feedback. Regarding this point, we also answered to reviewer 1, which we copied here:
We apologize for this, but we do not understand why the images were extremely low quality in our submitted version. Perhaps because we had to attach de pdf version instead of a word version. For us, at the submitted word version image quality is nice. To clarify image analysis and quality, we attached the three original high quality images. Also, we provided you a pdf version of the revised manuscript, hoping it will improve the images quality in this revised version of our manuscript
Actually, these imagens were not remade from cited literature. Figures 1 and 2 were based on some information provided by the following articles: [9,11,12,18,27,28], which were updated. Moreover, we also consolidated data that were scattered throughout the literature. Our aim was to try to correlate these data in some few figures, having the images from OPN genome sequences, their resulting transcripts, and then the protein sequences and corresponding their protein domains.
Figure 3, otherwise, is the original result of our brainstorm from data we collected from the literature regarding the information we presented at our review article. It did not used any previous published image information to be constructed.
